# Error Norm Truncation:
# Robust Training in the Presence of Data Noise for Text Generation Models

**Tianjian Li, Haoran Xu, Philipp Koehn, Daniel Khashabi$^{\heartsuit}$, Kenton Murray$^{\heartsuit}$**
Center for Language and Speech Processing
Johns Hopkins University, Baltimore MD
`{tli104, hxu64}@jhu.edu`

## Abstract

Text generation models are notoriously vulnerable to errors in the training data. With the wide-spread availability of massive amounts of web-crawled data becoming more commonplace, how can we enhance the robustness of models trained on a massive amount of *noisy* web-crawled text? In our work, we propose Error Norm Truncation (ENT), a robust enhancement to the standard training objective that truncates noisy data. Compared to methods that only use the negative log-likelihood loss over target words to estimate data quality, our method provides a more accurate estimation by considering the distribution of non-target tokens, which is often overlooked by previous work. Through comprehensive experiments across language modeling, machine translation, and text summarization, we show that equipping text generation models with ENT improves generation quality over standard training and previous soft and hard truncation methods. Furthermore, we show that our method improves the robustness of models against two of the most detrimental types of noise in machine translation, resulting in an increase of more than 2 BLEU points over the MLE baseline when up to 50% of noise is added to the data.

## 1 Introduction

Advances in neural text generation models have achieved remarkable success in various downstream tasks, which include but not limited to machine translation (Kalchbrenner & Blunsom, 2013), summarization (Rush et al., 2015), question answering (Joshi et al., 2017) and story generation (Fan et al., 2018). The prevalent paradigm of training text generation models is maximum-likelihood estimation (MLE), which finds parameters that maximize the probability of each token from the training data conditioned on a given context.

The limitation of MLE is that the model is forced to assign a non-zero probability to all tokens that appear in the training data, regardless of their quality, making the model not robust to errors in the training data. Existing research has demonstrated that text generation models are vulnerable to natural noise, such as misspelled and misordered words (Khayrallah & Koehn, 2018) and adversarial noise, such as poisoned training data (Wang et al., 2021a; Wallace et al., 2021; Wan et al., 2023).

To overcome this limitation, previous studies have either explored options to find alternatives to the autoregressive MLE paradigm (Khandelwal et al., 2021; Lewis et al., 2020b; An et al., 2022) or modify the MLE objective (Welleck et al., 2020; Li et al., 2020; Kang & Hashimoto, 2020; Lin et al., 2021; Pang & He, 2021; Xu et al., 2022; Ji et al., 2023). Modifications of MLE estimate data quality using the predicted probabilities of the ground truth token during training: a high probability corresponds to a higher likelihood that the ground truth token is clean and vice versa. Therefore, we can either directly remove data with high loss (Kang & Hashimoto, 2020; Goyal et al., 2022; Mohiuddin et al., 2022), or down-weigh data with low probability (Li et al., 2021; Ji et al., 2023) at each training iteration to improve robustness to data noise.

However, estimating data quality only using the predicted probability of the target token ignores the **distribution of the non-target tokens**. For example, when a model assigns a low probability to a

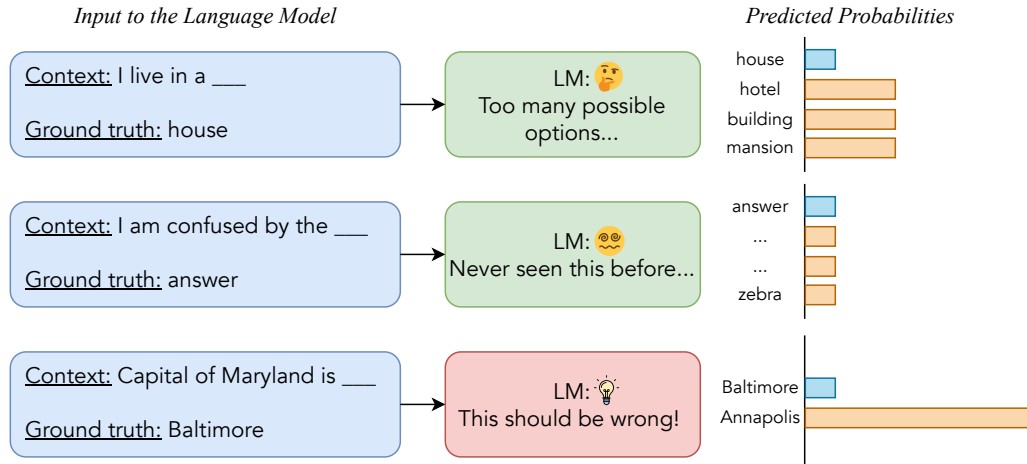

**Figure 1:** An motivating example of using the error norm for data quality estimation. All three examples have equal loss because they assign the same probability to the ground truth token. The skewness of the distribution of non-target tokens differentiates between the case when the context has high entropy with multiple possible continuations (example 1), when the model is at the beginning of training and is incompetent in making a prediction (example 2) and the case when the data is an error (example 3). **Truncating high loss removes all three examples whereas truncating high $\ell_2$ error norm only removes the third erroneous example.**

specific token, it could be the case that the context is high-entropy with many viable continuations, leading to a diluted probability of the target token (first example in Figure 1). Another possibility is that the model has not sufficiently converged and thus has not learned a reasonable distribution for this token (second example in Figure 1). In both cases, truncating this token or down-weighing the loss of this token could be harmful to model training.

To consider the predicted distribution of non-target tokens when estimating data quality, we propose **Error Norm Truncation** (ENT). This modified objective uses the $\ell_2$ norm of the difference between the model's predicted distribution and the one-hot vector of the ground truth to measure the quality of the data at each training iteration and truncate data with low quality. Intuitively, our method truncates tokens to which the model not only assigns a low probability but is very confident that it should be another token (third example in Figure 1). ENT improves robustness to data noise during training by accurately estimating data quality at the token level and removing noisy tokens.

To sum up, our contribution is threefold:

- We propose Error Norm Truncation: a data truncation method during training guided by a more accurate data quality estimation method that considers the probability distribution of non-target tokens;

- Through experiments under different tasks and setups, we show Error Norm Truncation consistently outperforms the MLE baseline as well as strong baselines proposed by previous methods in generation quality;

- We directly validate that Error Norm Truncation improves the robustness of machine translation models against two different types of noise: untranslated and randomly shuffled target sentences and outperforms all previous methods that truncate data.

## 2 BACKGROUND AND MOTIVATION

**Notation and Task Description.** We consider an conditional text generation model $p_\theta(\boldsymbol{y}|\boldsymbol{x})$. Given context $\boldsymbol{x}$ and target sequence $\boldsymbol{y} = (y_1, ..., y_T)$, the autoregressive framework models the probability of the target sequence conditioned on the context $p_\theta(\boldsymbol{y}|\boldsymbol{x})$ by factorizing it to the sum of log-probabilities of individual tokens. The prediction for each time step $t$ is conditioned both on the context $\boldsymbol{x}$ and the previous tokens $\boldsymbol{y}_{<t}$:

$$\log p_\theta(\boldsymbol{y}|\boldsymbol{x}) = \sum_{t=1}^{T} \log p_\theta(y_t|\boldsymbol{y}_{<t}, \boldsymbol{x}).$$

**Figure 2:** Examples of natural data noise that harms training. **Left**: summarization example from the XLSUM (Hasan et al., 2021) dataset where details in the summary (highlighted in red) cannot be inferred from the input text, which might cause the model to hallucinate facts in generating a summary. **Right**: Translation examples from opus-100 (Zhang et al., 2020), IWSLT 14 (Federico et al., 2014) and WMT 17 (Bojar et al., 2017), where details in the translation (highlighted in red) cannot be traced back to the source text (example 1 and 3), or requires the model to perform metric conversion (example 3).

The context $\boldsymbol{x}$ depends on the specific task: In machine translation, the context $\boldsymbol{x}$ is the source sentence to be translated from. In summarization, the context $\boldsymbol{x}$ is the article to be summarized. Standard language modeling can be seen as a special case where the context $\boldsymbol{x}$ is empty.

MLE maximizes the probability of the target sequences from a training corpus $\mathcal{D}$ by minimizing the expectation of the negative log-likelihood over the training corpus:

$$\mathcal{L}_\theta(\boldsymbol{x}, \boldsymbol{y}) = \mathbb{E}_{\boldsymbol{y} \sim \mathcal{D}} \left[ \sum_{t=1}^{T} - \log p_\theta(y_t | \boldsymbol{y}_{<t}, \boldsymbol{x}) \right].$$

However, the MLE objective is not robust to noise (Ji et al., 2023), which can be observed by calculating the gradient of the MLE loss function with respect to a single token $y_t$:

$$\nabla \mathcal{L}_\theta(\boldsymbol{x}, y_t) = - \frac{\nabla p_\theta(y_t | \boldsymbol{y}_{<t}, \boldsymbol{x})}{p_\theta(y_t | \boldsymbol{y}_{<t}, \boldsymbol{x})}.$$

When the data is incorrect and the predicted probability for the token $y_t$ (the denominator) is very small, the gradient norm $\|\nabla \mathcal{L}_\theta(x, y_t)\|$ would be very large, resulting in a large gradient update to an undesired direction.

**Previous Works.** The vulnerability of the MLE objective to noise cultivates research into truncating noisy data. A trivial method of estimating data quality $q(\boldsymbol{x}, \boldsymbol{y})$ is to use the predicted probability $p_\theta(\boldsymbol{y}|\boldsymbol{x})$. Intuitively, if the model assigns a low prediction probability to a training instance, it is more likely that the training instance is of low quality. However, in practice, a low prediction probability can also indicate a high entropy context rather than data quality.

A natural way to mitigate this vulnerability is to hard remove the noisy data: **Loss Truncation** (Kang & Hashimoto, 2020) directly removes a fixed fraction of the training **sentences** with the highest loss by setting their loss to 0, given a fraction of data $c$ to prune out. The loss function for Loss Truncation is:

$$\mathcal{L}_{\text{LT}} = - \log p_\theta(\boldsymbol{y}|\boldsymbol{x}) \cdot \mathbb{1}\big(p_\theta(\boldsymbol{y}|\boldsymbol{x}) > \tau_{\theta,c}\big),$$

where $\mathbb{1}(\cdot)$ is the indicator function and $\tau_{\theta,c}$ is the threshold calculated by the $c$-th percentile of losses over the training data. Note that the threshold depends on the model's current state since we use the model to rank training data and prune out a given percentage with the highest loss (or lowest predicted probabilities).

Data truncation can also be done in a soft and fine-grained way: **TaiLr** (Ji et al., 2023) up-weighs **individual tokens** with higher predicted probabilities, smoothed by an interpolation between the

ground truth distribution and the predicted probability of the model. The loss function $\mathcal{L}_{\text{TaiLr}}$ is:

$$\mathbb{E}_{\boldsymbol{y}\sim\mathcal{D}}\left[-\sum_{t=1}^{T}\underbrace{\left(\frac{p_\theta(y_t|\boldsymbol{y}_{<t},\boldsymbol{x})}{\gamma+(1-\gamma)\cdot p_\theta(y_t|\boldsymbol{y}_{<t},\boldsymbol{x})}\right)}_{\text{Weighting Factor}}\cdot\underbrace{\log p_\theta(y_t|\boldsymbol{y}_{<t},\boldsymbol{x})}_{\text{Standard Loss}}\right],$$

where $\gamma$ is a hyper-parameter for the smoothing factor. To overcome the issue of the model assigning a very small probability to all target tokens uniformly during the initial stage of training, TaiLr sets a lower threshold on the weighting factor as a hyperparameter. In our work, we consider Loss Truncation and TaiLr the most important baselines to compare.

**Motivation.** We point out two limitations of estimating data quality only by training loss:

- It is sensitive to the training iteration at which we start to estimate data quality and remove or down-weigh low-quality data.

- It ignores the rich information contained in the probability distribution of the incorrect (non-target) tokens, treating high and low entropy contexts as equal.

The first limitation arises from the model, when trained from scratch, undergoes multi-rounds of memorizing and forgetting (Toneva et al., 2019; Jiang et al., 2021; Jagielski et al., 2023) of individual examples. When a certain example is memorized, the model would label it as high quality and vice versa. This leads to high variance in measuring data quality throughout different stages of training. To overcome this issue, Loss Truncation first trains the model for a pre-defined number of iterations and then uses it to do quality estimation. TaiLr uses a pre-defined lower bound on the weighting factor. However, these methods require extensive hyper-parameter tuning due to the high variance, especially when estimating quality within a mini-batch at an arbitrary training iteration.

The second limitation arises from negative log-likelihood loss ignores the skewness of the probability distribution over non-target tokens. For example, when the model assigns a low probability to the ground truth token 'house', it might have distributed the majority amount of probability mass to synonyms 'building', 'hotel' and 'mansion'. There exist multiple correct predictions for a given context (Ott et al., 2018; Khayrallah et al., 2020), and only using the probability of one token to indicate quality leads to misjudgment.

**Figure 3:** The training dynamics of pre-training GPT2-large on WikiText-103. The plot shows the error norm for the largest 10% of data in each mini-batch. Initially, all error norms are close to 1, indicating the model uniformly assigns tiny probabilities to all target tokens. After the model is warmed up, it begins to detect data noise by assigning large error norms.

## 3 ERROR NORM TRUNCATION

Motivated by methods in dataset pruning (Paul et al., 2021), we propose to estimate data quality using the $\ell_2$ norm of the difference vector between the model's predicted distribution $p_\theta(\cdot|\boldsymbol{y}_{<t},\boldsymbol{x})$ and the groundtruth one-hot distribution $\text{OH}(y_t)$:

$$q(y_t,\boldsymbol{x}) = \|p_\theta(\cdot|\boldsymbol{y}_{<t},\boldsymbol{x}) - \text{OH}(y_t)\|_2,$$

which we refer as the **error norm**. $\text{OH}(y_t)$ is a vector with all zeros except the entry at $y_t$ is one. At each training iteration, we set a threshold as a hyper-parameter and hard prune out the tokens with an error norm above the threshold. The loss function for Error Norm Truncation (ENT) is:[1]

$$\mathcal{L}_{\text{ENT}} = \mathbb{E}_{\boldsymbol{y}\sim\mathcal{D}}[-\log p_\theta(\boldsymbol{y}|\boldsymbol{x})\cdot\mathbb{1}(q(\boldsymbol{y}_t,\boldsymbol{x}) < \tau_{\theta,c})].$$

The $\ell_2$ error norm presents a solution jointly to the two aforementioned limitations due to an observation: **the probability distribution of the incorrect tokens only becomes skewed after multiple**

---

[1]We provide PyTorch style pseudocode of Error Norm Truncation in Appendix D.

**iterations of training**. Initially, when the model does not have enough knowledge to make a prediction, the error norm for all data is close to 1, indicating that our model uniformly assigns probabilities to all target tokens. After multiple iterations of training, when the model has enough knowledge, the error norm of data noise becomes significantly larger. Figure 3 illustrates the state transition of the model from warming up to being able to make an estimate of data quality, corresponding to the horizontal red line at around training iteration 500. Setting a threshold on error norm allows the model to learn from all the data during the initial stage to make an educated estimate of data quality.

**Theoretical Connections.** As Kang & Hashimoto (2020) points out, a measurement of difference between probability distributions that is more robust to noise than the standard KL-Divergence (KLD) Kullback & Leibler (1951) is the Total Variation Distance (TVD) (van Handel, 2016), defined by the supremum of difference assigned to the same event. Intuitively, TVD measures the distinguishability between two distributions. Given two probability distributions $p$ and $q$ over all possible sequence $\mathcal{Y}$, the TVD between them is:

$$\text{TVD}(p, q) = \sup_{\boldsymbol{y} \in \mathcal{Y}} |p(\boldsymbol{y}) - q(\boldsymbol{y})|.$$

Ji et al. (2023) factorizes the sequence level TVD to the token level and proves that the token level TVD is an upper bound of the sequence level TVD, therefore minimizing the token-level TVD is able to make the model more robust to noise in the data. We show connections between error $\ell_2$ norm, the token-level TVD and the KL-Divergence.[2] By Pinsker's Inequality, we have

$$\underbrace{\frac{1}{2} \|p_\theta - \text{OH}(y_t)\|_2}_{\text{Error } \ell_2 \text{ Norm}} \le \frac{1}{2} \|p_\theta - \text{OH}(y_t)\|_1 = \underbrace{\sup_{y \in \mathcal{V}} |p(y) - \text{OH}(y_t)|}_{\text{Estimator of Token TVD}} \le \sqrt{\frac{1}{2} \text{KLD}(p_\theta \| \text{OH}(y_t))}.$$

We see that the error $\ell_2$ norm is a lower bound of the estimator of token level TVD. Examples with high error norm indicate a higher total variation distance, whereas examples with high loss (KLD) do not necessarily indicate a high TVD since it is a loose (Canonne, 2023) upper bound. Therefore, truncating examples with high error norms removes noisy data that has a higher TVD with the model's prediction learned from other instances.

## 4 CASE STUDIES

**Error Norm clearly distinguishes between clean and noisy tokens.** It is well established in robust statistics that $\ell_2$ error norm is more sensitive to outliers (Hastie et al., 2001) than $\ell_1$ norm, so $\ell_2$ norm is better in detecting outliers in data than $\ell_1$ norm. We prove the equivalency of using the error $\ell_1$ norm and standard loss in ranking data quality at Appendix A. To empirically show the superiority of using the $\ell_2$ norm in distinguishing between clean and noisy tokens, we use the dataset from Kang & Hashimoto (2020) which contains 300 examples from the Gigaword text summarization dataset where each summary is annotated into two categories: 1) directly entailed and 2) contains facts that cannot be inferred from the context. We find the precise tokens that are not entailed by the input and label them as `hallucinate` and label all the other tokens as `clean`.

We plot the normalized histograms of negative log-likelihood loss and error norm between clean and hallucinate tokens at figure 4a and 4b, evaluated by a pre-trained BART-large model. The overlap between clean and noisy distributions of loss (shaded area in figure 4a) is larger than the overlap of error norm (shaded area in figure 4b), indicating that error norm distinguishes between clean and noisy examples more clearly than negative log-likelihood loss.

**Error Norm provides a more accurate measure of data quality.** We directly verify that our method does provide a more accurate estimate of data quality. We plot out the BLEU scores of multilingual machine translation of 4 directions: En={De, Fr, It, Es} with a fixed fraction of **sentences** pruned out according to different metrics at Figure 5. ENT was able to match the performance of the baseline at small pruning fractions (10%-20%) while having in the least drop of performance

---

[2]For simplicity, we rewrite the probability distribution of predicted probabilities $p_\theta(\cdot | \boldsymbol{y}_{<t}, \boldsymbol{x})$ as $p_\theta$.

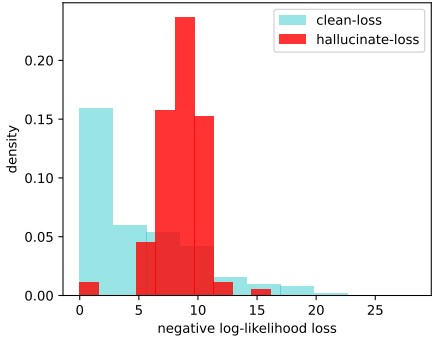

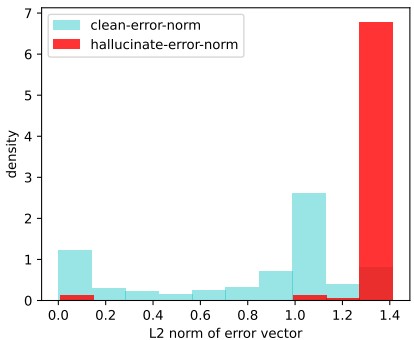

**(a)** Normalized histograms of log-likelihood loss.  **(b)** Normalized histograms of error norm.

**Figure 4:** Distributions of negative log-likelihood loss and error $\ell_2$ norm of clean and noisy data, evaluated by a pre-trained BART-large model. Error norm clearly distinguishes between clean and noisy data.

at high pruning fractions, outperforming randomly pruning for 2.43 BLEU and outperforming Loss Truncation by 0.88 BLEU when 60% of the data is pruned out. This shows that Error Norm provides a more accurate estimate of data quality than negative log-likelihood loss.

## 5 EXPERIMENTS

In this section, we show that truncating tokens with high error norm improves generation quality across different tasks. We describe the setup for all of our experiments at §5.1. We validate that our methods improves robustness under synthetic noise at §5.2. We present our experiment results under the train-from-scratch setting at §5.3 and under the fine-tune setting at §5.4. We include results of both truncating a fixed fraction of data (ENT-Fraction) and truncating according to a pre-defined threshold (ENT-Threshold). Detailed dataset statistics and hyper-parameters are at Appendix C.

### 5.1 SETUP

**Robustness Experiments.** To directly verify the ENT improves robustness, we inject noise into 1M parallel sentences of En-Fr data from the opus-100 dataset. We select two of the most harmful type of noise (Khayrallah & Koehn, 2018): **Untranslated Text** where the source sentence is directly copied to the target side; **Misordered Words** where the words at the target side is randomly shuffled. We vary the amount of noise added to the corpus {10%, 20%, 30%, 40% 50%} of the size of the original clean corpus and report the BLEU scores of models trained on MLE equipped with Loss Truncation, TaiLr and ENT-Fraction on the perturbed datasets.

**Train-from-Scratch.** We evaluate our method on machine translation and general language modeling. For multilingual translation, we train a single model for eight directions en-{es,fa,fr,it,ko,ru,tr,zh} from the opus-100 cor-

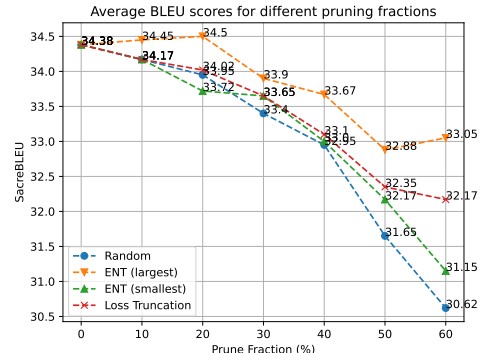

**Figure 5:** Average BLEU results of 4 translation directions En-{De, Fr, It, Es} from the opus-100 dataset with a fraction of sentences being truncated according to loss, error norm, and randomly truncated. **Truncating high error norm sentences achieves the best performance at all truncation fractions.**

pus[3] (Zhang et al., 2020) using 1M parallel sentences for each direction.

We train on the fairseq (Ott et al., 2019) implementation of the standard Transformer (Vaswani et al., 2017) architecture [4] for all of our machine translation experiments. For language modeling, we train

---

[3]https://opus.nlpl.eu/opus-100.php
[4]`transformer_iwslt_de_en`

a GPT2-large (Radford et al., 2019) model on the WikiText-103 dataset (Merity et al., 2017) for 5 epochs from scratch. We use the Huggingface (Wolf et al., 2020) implementation of GPT2-large.

**Fine-Tuning.** We validate our method on the text summarization CNN/Daily Mail (See et al., 2017; Hermann et al., 2015) dataset on two different models: T5-small (Raffel et al., 2020) and BART-base (Lewis et al., 2020a) to validate our method generalizes across different pre-trained models. We use the Huggingface implementations of T5 and BART.

## 5.2 ROBUSTNESS RESULTS

**Untranslated Text.** Table 1 shows the BLEU results of machine translation models trained on corpus with different level of untranslated text injected. Since the corpus is high-quality data from the opus-100 training set, the difference between various methods that aim to improve robustness to noise is small when no noise is added.

The MLE baseline model's scores gradually decrease with increased injection, revealing the negative impact of untranslated sentences. Loss Truncation maintains similar BLEU scores. TaiLr exhibits modest gains in both metrics. Notably, Error Norm Truncation consistently improves performance with higher injection percentages. Outperforming the baseline 3.8 BLEU and outperforming the best of Loss Truncation and TaiLr 2.1 BLEU when 50% of noise is injected. These results emphasize the challenge of handling untranslated content, with the Error Norm Truncation proving exceptionally effective in mitigating this issue and enhancing translation quality.

**Misordered Words.** Table 2 shows the BLEU results of models when trained on data with misordered sentences injected at the target side. Our results echos with the results in Khayrallah & Koehn (2018), showing that randomly shuffling the target sentence is a weaker type of noise compared to directly copying the source text to the target. Although Loss Truncation was able to improve upon the baseline when a small amount of noise is added (10-20%), it performs the same as standard MLE training at when a larger amount of misordered sentences are added to the training data. ENT is the most resilient method against misordered words at the target side, resulting in the largest BLEU scores improvement over the baseline in all noise levels. It outperforms the baseline 0.9 BLEU when 50% of randomly shuffled sentences are injected and only underperforms 0.1 BLEU against the performance of standard training on clean data, indicating the resilience of the model against randomly shuffled target sentences when equipped with ENT.

| Untranslated | 0% | 10% | 20% | 30% | 40% | 50% |
|---|---|---|---|---|---|---|
| MLE | 36.5 | **34.9** | 33.2 | 30.6 | 31.0 | 28.6 |
| Loss Trunc. | 36.5 | 33.2 | 32.5 | 31.5 | 31.4 | 29.4 |
| TaiLr | 36.6 | 34.3 | 33.4 | 31.5 | 31.6 | 30.3 |
| ENT-Fraction | **36.7** | 33.3 | **33.8** | **33.3** | **33.1** | **32.4** |

**Table 1:** BLEU scores of models trained on opus-100 En-Fr data injected with the source sentence directly copied to the target side (Untranslated Text) ranging from 10% to 50% of the original clean data. Truncating with error norm is the most robust method against untranslated sentence.

| Misordered | 0% | 10% | 20% | 30% | 40% | 50% |
|---|---|---|---|---|---|---|
| MLE | 36.5 | 36.1 | 36.1 | 36.2 | 35.8 | 35.5 |
| Loss Trunc. | 36.5 | 36.1 | 36.1 | 36.2 | 35.8 | 35.7 |
| TaiLr | 36.6 | 36.2 | 36.2 | 36.3 | 36.2 | 36.2 |
| ENT-Fraction | **36.7** | **36.3** | **36.7** | **36.7** | **36.5** | **36.4** |

**Table 2:** BLEU scores of models trained on opus-100 En-Fr data injected with parallel sentences randomly shuffled (Misordered Words) at the target side ranging from 10% to 50% of the original clean data. Truncating with error norm was able to improve upon the baseline the most compared to existing methods.

## 5.3 TRAIN-FROM-SCRATCH RESULTS

**Language Modeling.** We first evaluate our method on general language modeling. Table 3 shows the results of the validation perplexity of pre-training a GPT-2 Large model on WikiText-103 from scratch. Hard truncation methods (Loss Truncation and Error Norm Truncation) were able to lower the perplexity by more than 1 point compared to the MLE baseline. Truncating with error norm

outperforms truncating with loss for a fixed fraction. Truncating to a given threshold outperforms all existing methods by lowering 1.58 perplexity compared to the MLE baseline.

|        | MLE   | Loss Truncation | TaiLr | ENT-Fraction | ENT-Threshold |
|--------|-------|-----------------|-------|--------------|---------------|
| PPL. ↓ | 25.88 | 24.64           | 25.62 | 24.50        | **24.30**     |

**Table 3:** Validation perplexity on WikiText-103 of pre-training a GPT2-large model with different data truncation methods. Truncating with error norm outperforms the MLE baseline by 1.38 perplexity while truncating to a given threshold further improves the performance by 0.2 points in perplexity.

To show that Error Norm Truncation is less sensitive to the iteration from which soft or hard data truncation methods are applied, we vary this iteration $\in \{0, 100, 200, 500, 1000\}$ parameter updates and plot out the validation perplexity on WikiText-103 of different methods at Figure 6. We see that ENT-Fraction is able to outperform previous methods while having the lowest variance and ENT-Threshold further improves the performance over ENT-Fraction. We highlight that large-scale language model pre-training is too expensive to tryout a combinatorically large number of hyper-parameters, therefore our method is more scalable to large-scale pre-training tasks compared to other methods due to the low variance and high performance.

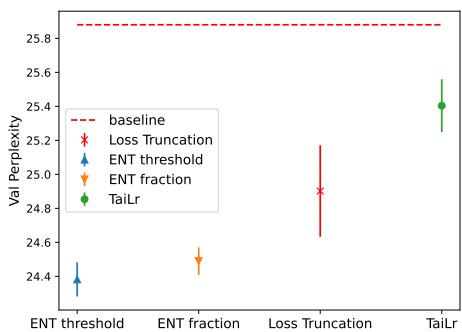

**Figure 6:** Validation perplexity↓ on WikiText-103 by varying the iteration to start using different methods. **ENT exhibits the least variance and best performance.**

**Machine Translation.** Table 4 shows the BLEU results on Multilingual Machine Translation, where 1M parallel sentences for each language pair from a set of linguistically diverse languages are concatenated for training a large model. We find that previous methods often underperform the MLE baseline due to not capturing the model's competency during truncating, while our method consistently outperforms the baseline. Our method also outperforms Loss Truncation in 6 out of 8 directions, given a fixed pruning threshold.

| En-{}           | Es    | Fa    | Fr    | It    | Ko    | Ru    | Tr    | Zh    | Avg.  |
|-----------------|-------|-------|-------|-------|-------|-------|-------|-------|-------|
| MLE             | 40.5  | 14.2  | 40.4  | 35.1  | 10.1  | 36.3  | 25    | 39.2  | 30.1  |
| Loss Truncation | 39.8  | 14.0  | 40.1  | 34.4  | 9.9   | 36.5  | 24.7  | **40.1** | 29.9  |
| TaiLr           | 40.4  | 14.0  | 40.2  | 35.1  | 10.0  | 36.1  | 25.2  | 39.6  | 30.1  |
| ENT-Fraction    | 41.1  | 14.8  | 40.3  | **35.2** | **10.3** | 36.4  | 25.0  | 39.6  | 30.3  |
| ENT-Threshold   | **41.9** | **14.9** | **41** | 34.8  | 10.2  | **36.5** | **25.5** | 39.8  | **30.6** |

**Table 4:** BLEU results on a linguistically diverse subset of the opus-100 dataset. **Error Norm Truncation with threshold and fraction outperforms the baseline and Loss Truncation in 7 out of 8 directions.**

## 5.4 FINE-TUNING RESULTS

**Summarization.** Table 5 shows the results of fine-tuning T5-small and BART-base on the CNN/-Daily Mail Summarization dataset. Since we can rely on the pre-trained model to make an estimate of the data quality, we do not need to pre-define a threshold for the model. Directly pruning out a fraction of data produces the best result in this case. Again, we were able to observe that truncating with error norm consistently outperforms all other methods in two different models.

## 6 RELATED WORKS

**Modifications to MLE for Text Generation.** As the MLE objective is not robust to noise, numerous work have proposed ways to modify the MLE objective. Welleck et al. (2020) proposes

|  | T5-small | | | BART-base | | |
|---|---|---|---|---|---|---|
|  | R-1 | R-2 | R-L | R-1 | R-2 | R-L |
| MLE | 42.19 | 19.69 | 39.04 | **43.50** | 20.59 | 40.36 |
| Loss Truncation | 42.22 | 19.68 | 39.05 | 43.22 | **20.66** | 40.44 |
| TaiLr | 41.53 | 19.22 | 38.33 | 42.20 | 19.66 | 39.07 |
| ENT-Fraction | **42.63** | **19.98** | **39.57** | 43.48 | 20.29 | **40.72** |
| ENT-Threshold | 42.37 | 19.80 | 39.27 | 43.35 | 20.30 | 40.54 |

**Table 5:** Best validation rouge-1/2/LSum results on fine-tuning T5-small and BART-base equipped with different robust modifications to MLE on the CNN/Daily Mail dataset. ENT is able to outperform baselines on T5-small and match the performance of baselines on BART-base.

to augment the MLE objective by penalizing the model for generating undesired outputs. Xu et al. (2022) directly penalizes the model for generating repetitions. Lin et al. (2021) modifies the gradient to encourage the model to generate diverse text. Kang & Hashimoto (2020) truncate a given fraction of data with the highest loss to remove noise from the data. Pang & He (2021) reformulates text generation as an off-policy and offline reinforcement learning problem, assigning weights to each token according to a pre-defined reward function. Similarly, Ji et al. (2023) also reweighs each token from the training dataset by the prediction probability of the model, smoothed by interpolation between the one-hot probability vector and the predicted probability vector. Li et al. (2020) points out that the standard MLE objective treats all incorrect tokens as equal and proposes to learn a prior distribution over the tokens using the training data and smooth the one-hot ground truth distribution to a Gaussian distribution over tokens with similar embeddings. Welleck et al. (2023) proposes first to generate an intermediate output using MLE and iteratively refines the generation. To the best of our knowledge, our work is the first to address the limitations of only relying on the output probabilities in estimating data utility.

**Measuring Data Utility in NLP.** Numerous works have proposed methods to estimate the contribution of each single data point in Natural Language Processing. For text generation tasks, the quality of data can be as simple as handcrafted heuristics such as word frequency and sequence length (Platanios et al., 2019), the relative position of the word in a sentence (Liang et al., 2021; Jia et al., 2023), the similarity to a target domain (Moore & Lewis, 2010; Zhang et al., 2019). Besides handcrafted heuristics, model generations (Wettig et al., 2024; Liu et al., 2024) and signals (loss, gradient, and representations) can also be utilized to measure data quality. Koh & Liang (2017) imports Influence Functions (Cook & Weisberg, 1975) from statistical theory to deep learning, measuring the utility of each training example by the difference between the parameters of the model trained with and without the particular training example. However, this estimation requires the computation of single sample gradients, which is impractical when the training dataset is large. Paul et al. (2021) shows that the influence on training loss of removing one particular training example is upper bounded by the gradient norm when trained on that example and proposes to approximate the single sample gradient norm by the error $\ell_2$ norm. All of the above methods assume that the data utility is static. Our work differs in that our method takes into account the training dynamics while making quality estimations. For a comprehensive survey on data selection for NLP, we refer the readers to Albalak et al. (2024). Additional related works on measuring data utility with model signals and discussions on Influence Functions are provided in Appendix B.

## 7 CONCLUSION AND LIMITATIONS

**Conclusion.** Our work proposes **Error Norm Truncation** (ENT), a robust modification to the standard MLE objective in training text generation models. ENT measures the quality of each token by considering the skewness of the predicted distribution and truncates the noisy tokens during training. ENT demonstrates enhanced stability and superior performance over existing methods.

**Limitations.** We acknowledge that the improvements of our method result from the noisy distribution of the training data, therefore the improvements on clean, curated data might not be as large. We leave more coarse-grained grouped data and dataset quality estimation for future work.

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

## A  EQUIVALENCE OF LOSS AND ERROR $\ell_1$ NORM

**Theorem:** Given datapoints $(\boldsymbol{x}_i, y_i)$ and $(\boldsymbol{x}_j, y_j)$, if $\mathcal{L}_\theta(\boldsymbol{x}_i, \boldsymbol{y}_{<i}, y_i) = \mathcal{L}_\theta(\boldsymbol{x}_j, \boldsymbol{y}_{<j}, y_j)$, then

$$\|p_\theta(\cdot \mid \boldsymbol{y}_{<i}, \boldsymbol{x}_i) - \mathrm{OH}(y_i)\|_1 = \|p_\theta(\cdot \mid \boldsymbol{y}_{<j}, \boldsymbol{x}_j) - \mathrm{OH}(y_j)\|_1.$$

Where $\mathrm{OH}(\cdot)$ is the one-hot vector.

**Proof:**

$$\mathcal{L}_\theta(\boldsymbol{x}_i, \boldsymbol{y}_{<i}, y_i) = \mathcal{L}_\theta(\boldsymbol{x}_j, \boldsymbol{y}_{<j}, y_j)$$
$$\implies p_\theta(y_i|\boldsymbol{y}_{<i}, \boldsymbol{x}) = p_\theta(y_j|\boldsymbol{y}_{<j}, \boldsymbol{x})$$
$$\implies 2 - 2 \cdot p_\theta(y_i|\boldsymbol{y}_{<i}, \boldsymbol{x}) = 2 - 2 \cdot p_\theta(y_j|\boldsymbol{y}_{<j}, \boldsymbol{x})$$
$$\implies |1 - p_\theta(y_i|\boldsymbol{y}_{<i}, \boldsymbol{x})| + \underbrace{1 - p_\theta(y_i|\boldsymbol{y}_{<i}, \boldsymbol{x})}_{\sum_{y \neq y_i} |p(y|\boldsymbol{y}_{<i}, \boldsymbol{x})|} = |1 - p_\theta(y_j|\boldsymbol{y}_{<j}, \boldsymbol{x})| + \underbrace{1 - p_\theta(y_j|\boldsymbol{y}_{<j}, \boldsymbol{x})}_{\sum_{y \neq y_j} |p(y|\boldsymbol{y}_{<j}, \boldsymbol{x})|}$$
$$\implies \|p_\theta(\cdot \mid \boldsymbol{y}_{<i}, \boldsymbol{x}) - \mathrm{OH}(y_i)\|_1 = \|p_\theta(\cdot \mid \boldsymbol{y}_{<j}, \boldsymbol{x}) - \mathrm{OH}(y_j)\|_1.$$

## B  ADDITIONAL RELATED WORKS

**Measuring Data Utility. Influence Functions** (Cook & Weisberg, 1975; Koh & Liang, 2017) measures the utility of data utilizing first and second order model signals (gradients and Hessian). Specifically, the score $q$ assigned to each training data pair $(\boldsymbol{x}, \boldsymbol{y})$, evaluated by model parameterized by $\theta$ is given by:

$$q(\boldsymbol{x}, \boldsymbol{y}) = -\nabla_\theta \ell(z_0; \theta)^\top \mathcal{H}_\theta^{-1} \nabla_\theta \ell(\boldsymbol{x}, \boldsymbol{y}; \theta)$$

where $z_0$ is the domain on which you want to evaluate your data utility. For standard training where you care about the influence on generalizability, $z_0$ is the test set. For domain adaptation, $z_0$ is data from the target domain. $\mathcal{H}_\theta^{-1}$ is the inverse Hessian.

Most of the work utilizing model signals for estimating data utility can be viewed as simplifications to Influence Functions. A line of work (Wang et al., 2020a; Yu et al., 2020; Wang et al., 2020b; Yang

et al., 2021; Wang et al., 2021b; Fan et al., 2024) drops the Hessian dependency and measures the data utility by the gradient similarity to the development set $q(\boldsymbol{x}, \boldsymbol{y}) = -\nabla_\theta \ell(z_{\text{dev}}; \theta)^\top \nabla_\theta \ell(\boldsymbol{x}, \boldsymbol{y}; \theta)$. Since the test set distribution should be unknown and relying on gradient similarity to the dev set risk overfitting to the dev set, another line of work (Pruthi et al., 2020; Paul et al., 2021) only uses the gradient norm $q(\boldsymbol{x}, \boldsymbol{y}) = \|\nabla_\theta \ell(\boldsymbol{x}, \boldsymbol{y}; \theta)\|$ in estimating the utility of data. Our work also falls into this category by approximating the gradient norm with the error vector norm and treating data utility as adaptive to the model competence rather than a fixed value.

Besides simplifying the Influence Function utility estimation, Basu et al. (2021) finds that the accuracy of Influence Function heavily depends on inductive biases and can break if the neural network is too deep. Koh et al. (2019) and Yang et al. (2023) extends beyond quantifying data utility of single examples by considering the interaction when multiple training instances are collectively pruned. Ladhak et al. (2023) trains the same model for one iteration on clean data and on noise, and use the difference in loss for finding errors in the training dataset, which can be seen as a realization of the gradient similarity between training on clean and noisy examples. Grosse et al. (2023) approximates the inverse Hessian in the influence function using the Eigenvalue-corrected Kronecker-Factored Approximate Curvature (EK-FAC) and batch similar queries together to overcome the bottleneck of computing single sample gradients. Besides truncating data to improve robustness, data utility measuring with Influence Functions can also be applied to understanding model generalization (Grosse et al., 2023), explaining black-box predictions (Han et al., 2020), finding spurious correlations (Han & Tsvetkov, 2021), and studying the impact of label errors on model disparity metrics (Adebayo et al., 2023).

**Active Learning and Uncertainty Sampling.** Active learning aims to select the most informative data for labeling within a given annotation budget. Uncertainty sampling, as an active learning algorithm, targets datapoints where the model exhibits the highest uncertainty. The two simplest techniques for uncertainty sampling, as outlined by Weng (2022), are:

- Loss: Selecting datapoints with the lowest predicted probabilities $p_\theta(\hat{y}|\boldsymbol{x})$,
- Entropy: Selecting datapoints with high entropy $-\sum_y p_\theta(y|\boldsymbol{x}) \log p_\theta(y|\boldsymbol{x})$.

Utilizing loss for data selection is connected to Loss Truncation (Kang & Hashimoto, 2020). The distinction lies in the fact that instead of truncating high-loss examples, uncertainty sampling opts to train on such challenging instances, allowing the model to focus on handling difficult cases.

The selection of high-entropy data is associated with employing the $\ell_2$ norm of the model's prediction probability vector $\|p_\theta(\cdot|\boldsymbol{x})\|_2$. Rényi (1961) establishes the equivalence between selecting data with high Rényi entropy and selecting data with a low $\ell_2$ norm of the predicted probability vector:

$$H_2(p_\theta(\cdot|\boldsymbol{x})) = -\log\left(\|p_\theta(\cdot|\boldsymbol{x})\|_2\right).$$

ENT combines the benefits of both loss and entropy-based data selection by using the $\ell_2$ norm of the error vector: $\|p_\theta(\cdot|\boldsymbol{x}) - \text{OH}(\hat{y})\|_2$. Data with a high error $\ell_2$ norm comprises instances with low predicted probability and low entropy. Intuitively, ENT truncates data that the model is certain is incorrect.

## C  Tasks, Model Sizes, and Hyper-Parameters

Table 6 shows the datasets, sizes and the evaluation metrics that we used in our paper.

**Hyper-parameters.** We use the official implementation[5] of Loss Truncation and re-implement TaiLr ourselves. For a fair comparison with Loss Truncation, we include results of both truncating a fixed fraction of data (ENT-Fraction) and truncating according to a pre-defined threshold (ENT-Threshold). We fix the truncation fraction to be 0.1 for ENT-fraction and choose the best result among three truncation fractions $\{0.05, 0.1, 0.2\}$ for Loss Truncation. For TaiLr, in addition to the recommended hyperparameter setting for machine translation and summarization in Ji et al. (2023), we additionally tuned $3 \times 3$ hyperparameter combinations: $\gamma \in \{0.1, 0.5, 1.0\}$ and lower threshold of the weighting factor among $\{0.1, 0.2, 0.3\}$. We select the best results among three threshold values

---

[5]https://github.com/ddkang/loss_dropper

| | Dataset | Size | Metric |
|---|---|---|---|
| | *Trained-from-Scratch* | | |
| Bilingual MT | ParaCrawl | En-Cs (55M), En-Ru (50M), En-Zh (13M) | SacreBLEU |
| Multilingual MT | opus-100 | 1M sentences for all directions | SacreBLEU |
| Multilingual MT | opus-100 | En-Gl (400K), En-Fr (1M) | SacreBLEU |
| Language Modeling | WikiText-103 | 100M tokens | Perplexity |
| | *Fine-tuning* | | |
| Summarization | CNN/Daily Mail | 286K article-summary pairs | Rouge-1/2/LSum |
| | *Robustness* | | |
| Bilingual MT | opus-100 | En-Fr (1M) | SacreBLEU |

**Table 6:** Dataset statistics for our experiments. We report the number of parallel sentences for all machine translation experiments.

$\{1.35, 1.38, 1.4\}$ for ENT-threshold.[6] For all of our Machine Translation experiments, we report SacreBLEU (Post, 2018) results[7] on the test set. For all of our experiments, we report the average results of three runs with different random seeds.

## D ALGORITHM PSEUDOCODE

**Algorithm 1:** Error Norm Truncation - Fraction

```
# Input:
# logits:  torch.Tensor, the output logits from the LM.
# shape of (batch size, seq length, vocab size)
# labels:  torch.Tensor, the one-hot vector of target tokens
# shape of (batch size, seq length, vocab size)
# fraction:  float, the fraction of tokens to prune.
#
# Output:
# Loss:  torch.Tensor.
#
# Compute binary mask
probs = nn.functional.softmax(logits, dim=-1)
en = torch.linalg.norm(probs - labels dim=-1)
sorted_en = torch.sort(en.view(-1), descending=True).values
threshold = sorted_en[int(fraction * len(sorted_en))]
# threshold ← fixed number < sqrt(2) in ENT-Threshold
mask = en > threshold
# Compute loss
loss_fn = nn.NLLLoss(reduction='none')
loss = loss_fn(torch.log(probs), target)
loss = loss.mean()
```

## E EXAMPLES

Table 7 shows examples from the opus-100 dataset where errors are found by large error norms.

---

[6]The threshold values was based on preliminary experiments: the maximum of error $\ell_2$ norm is $\sqrt{2} \approx 1.414$

[7]BLEU|nrefs:1|case:mixed|eff:no|tok:flores200|smooth:exp

| | |
|---|---|
| Source: <2fr>They are used to forcast cereals, industrial and other crops. | |
| Target: Elles sont utilisées pour les prévisions concernant `les` `cultures` céréalières, industrielles et autres. | |

| | |
|---|---|
| Source: <2fr>And make me of the heirs of the garden of bliss. | |
| Target: `et` fais de moi l ' un des héritiers du `J` ardin `des` délices. | |

| | |
|---|---|
| Source: <2de>Look for me in the end zone after this play. | |
| Target: `Red 7, Red 7, Red 7` ! Du `find est` mich nach dem Spiel in der End `-Zone` . | |

**Table 7:** Translation examples from the opus-100 dataset. Tokens with error norm larger than 1.0 are highlighted in yellow and tokens with error norm larger than 1.3 are highlighted in red. The error norm helps us spot mistakes in the data. Instead of removing entire sentences, focusing on the highlighted tokens for truncation preserves the rest of the sentence, which can still hold valuable information.

## F   BILINGUAL MACHINE TRANSLATION RESULTS

For bilingual translation, we train seperate models for the following three directions en-{cs,ru,zh} from the ParaCrawl V9 corpus[8] (Bañón et al., 2020) and report the BLEU (Papineni et al., 2002) results on the WMT22 test set (Kocmi et al., 2022).

Table 8 shows the BLEU scores of equipping MLE with error norm truncation compared with other soft and hard truncation baselines. ENT-fraction outperforms Loss Truncation in all three directions. ENT-Threshold is able to outperform all previous methods in directions En-Cs and En-Ru, only behind the best performance of En-Zh by 2 BLEU points.

| | En-Cs | En-Ru | En-Zh |
|---|---|---|---|
| MLE | 25.2 | 24.6 | 12.5 |
| Loss Truncation | 25.2 | 25.3 | 12.8 |
| TaiLr | 25.1 | 25.4 | **13.2** |
| ENT-Fraction | 25.3 | **25.5** | 13.1 |
| ENT-Threshold | **25.7** | **25.5** | 13 |

**Table 8:** Monolingual Machine Translation BLEU results trained on the ParaCrawl dataset and evaluated on WMT22 test set. Error Norm Truncation outperforms the baseline and other data truncation methods.

## G   MULTILINGUAL MACHINE TRANSLATION WITH MISMATCHED DATA SIZES

Table 9 shows the multilingual machine translation results when there is a mismatch in data size. Error norm truncation improves more on the low resource language pair En-Gl more compared to the improvements on the high resource language in all 3 temperature settings, indicating that removing noisy data can balance training in under a mismatched multilingual setting, improving the performance on low-resource languages without sacrificing performance on high-resource languages.

---

[8]https://statmt.org/wmt22/translation-task.html

| | T=1 | | T=5 | | T=100 | |
|---|---|---|---|---|---|---|
| | En-Gl | En-Fr | En-Gl | En-Fr | En-Gl | En-Fr |
| MLE | 27.4 | 38.1 | 27.1 | 37.2 | 27.9 | 37.2 |
| Loss Truncation | 27.4 | 37.9 | 27.3 | 37.0 | 27.6 | 37.1 |
| TaiLr | 27.7 | 38.0 | **27.5** | 37.1 | 28.2 | **37.5** |
| ENT-Fraction | 28.0 | **38.2** | 27.4 | 37.2 | 28.2 | 37.2 |
| ENT-Threshold | **28.1** | **38.2** | **27.5** | **37.3** | **28.5** | 37.2 |

**Table 9:** BLEU results of multilingual machine translation under 3 different sampling temperatures. Our method was able to outperform the baseline and other truncation methods in 5 out of 6 setups. En-Gl is low resource with 400k parallel sentences and En-Fr is high resource with 1M parallel sentences.

