# OpenReview forum: "Error Norm Truncation: Robust Training in the Presence of Data Noise for Text Generation Models"
_ICLR.cc/2024/Conference — ICLR 2024 spotlight_

### Official Review · Reviewer_p8xo · 2023-11-02

**Soundness:** 3 good
**Presentation:** 3 good
**Contribution:** 2 fair
**Rating:** 6
**Confidence:** 4

**Summary:**

This paper proposes a method for automatically identifying which data points are potentially noisy.
Specifically, this is done based on identifying points that have high L2 norms of the error and removing them from the loss function.

**Strengths:**

* Overall, I do really like the concept of the paper. It is simple and generally makes sense.
* The method seems to have strong empirical results in synthetic settings, and reasonable experimental results in less synthetic settings.
* Figure 4 is a nice analysis demonstrating that intuitively why the proposed method is better than log likelihood.

**Weaknesses:**

I have a few concerns about the paper:

* It seems that some hyperparameter tuning (detailed in appendix C) was done for all of the loss modification methods, but none was done for MLE. Because of this, perhaps some of the gains over MLE can be attributed to randomness in training, rather than to inherent goodness of the method.
* The results on real-world datasets (sections 5.3 and 5.4) are somewhat underwhelming. I see small gains (and perhaps small gains are a good result already given how simple the method is), but I'm also a little bit concerned whether these are interesting enough for practitioners to be excited and go back and implement/use this method. Overall, I feel like the paper lacks a big convincing results, such as  significant improvements to SOTA on a dataset that people care about.

Note that I am not saying that the work is solid, it seems to be done reasonably well, I'm just not sure how much impact it will have on the community given the current empirical evidence.

**Questions:**

1. How was the thereshold hyperparameter tuned in all experiments in the experimental section?
2. I was confused by the second equation in section 3, should it be a "less-than" sign rather than a "greater-than" sign?

---

> ### Author Response · Authors · 2023-11-19
> **Response to reviewer p8xo**
>
> We sincerely thank the reviewer for the insightful feedback and for appreciating the simplicity of our method, as well as for acknowledging our “strong empirical results in synthetic settings, and reasonable experimental results in less synthetic settings.”
>
> We hope to address the concerns of the reviewer below:
>
> > perhaps some of the gains over MLE can be attributed to randomness in training, rather than to inherent goodness of the method.
>
> To address the reviewer’s concerns, we performed additional tuning on the MLE baseline for our opus-MT experiments (Table 4), varying the learning rate and dropout probability:
>
> | lr              | dropout | BLEU |
> |-----------------|---------|------|
> | 5e-4 (our reported baseline) | 0.1     | 30.1 |
> | 1e-3            | 0.1     | 30.0 |
> | 1e-4            | 0.1     | 28.4 |
> | 7e-4            | 0.1     | **30.2** |
> | 5e-4            | 0       | 29.1 |
> | 5e-4            | 0.2     | **30.2** |
> | 5e-4            | 0.3     | 29.8 |
>
> **We tuned the baselines extensively.** We show the number of additional hyper-parameter configurations for each baseline in the following chart:
>
>  | Method         | Number of Hyper-parameters Tuned |
> |---------------------|----------------------------------|
> | TaiLr [3]           | 10                               |
> | Loss Truncation [2] | 3                                |
> | ENT-Threshold       | 3                                |
> | ENT-Fraction        | 1                                |
>
> > Overall, I feel like the paper lacks a big convincing results, such as significant improvements to SOTA on a dataset that people care about.
>
> Our main claim is that our method improves the robustness of text generation models against various types of data noise, outperforming **existing truncation methods based only on loss** across three different tasks, rather than our method achieving SOTA in language generation. SOTA methods for different tasks usually require task-specific handling (e.g., using bilingual pre-trained masked language models for translation [1], using contrastive learning between candidate summaries for summarization [4]), making them less generalizable. In contrast, our proposed approach **does not make such task-specific assumptions**, making it applicable to any text generation task to improve robustness.
>
> Moreover, **our gains are not limited to synthetic settings**: We would like to highlight that our method yields large improvements in language modeling (-1.58 Perplexity, Table 3), and in Appendix F, we show that our method improves upon the MLE baselines from +0.5 to 0.9 BLEU on Paracrawl MT data. To further address the reviewer’s concerns, we added experiments to show that our method can be used in conjunction with SOTA task-specific methods [1, 4] to improve the performance:
>
> | IWSLT 14 De-En  | Test BLEU |
> |----------------------------------|-----------------|
> | BiBERT (Table 2 in [1])      | 37.58     |
> | BiBERT (reproduced) | 37.62     |
> | BiBERT + ENT-Fraction        | **38.29**    |
>
> | CNN/DM            | Rouge-1 | Rouge-2 | Rouge-L |
> |-------------------|---------|---------|---------|
> | BRIO (Table 2 in [4])   | 47.78   | 23.55   | 44.57   |
> | BRIO (reproduced) | 47.81   | 23.52   | 44.52   |
> | BRIO + ENT-Fraction  | **48.04**   | **23.61**   | **44.81**   |
>
> We also conducted additional experiments in a more realistic setting: We use the Human+GPT mixture data in [5] and perform instruction tuning on LLama 2 7B. We show the results of adding ENT below:
>
> |                                   | MMLU (0-shot Acc.) | TyDiQA (F1) |
> |-----------------------------------|--------------------|-------------|
> | Llama2-7b instruct (Table 5 in [5])     | 49.2               | 52.8        |
> | LLama2-7b instruct (reproduced)   | 50.1               | 52.1        |
> | LLama2-7b instruct + ENT-Fraction | **51.3**               | **54.0**        |
>
>
> > How was the threshold hyperparameter tuned in all experiments in the experimental section?
>
> We choose the best of {1.35, 1.38, 1.4} for ENT-Threshold in all our experiments. More details of how we tuned the hyper-parameters can be found in Appendix C.
>
> >  I was confused by the second equation in section 3, should it be a "less-than" sign rather than a "greater-than" sign?
>
> We thank the reviewer for pointing this out, and we have made revisions to the manuscript (marked in blue).
>
> [1] BERT, mBERT, or BiBERT? A Study on Contextualized Embeddings for Neural Machine Translation (Xu et al., EMNLP 2021)
>
> [2] Improved Natural Language Generation via Loss Truncation (Kang & Hashimoto, ACL 2020)
>
> [3] Tailoring Language Generation with Total Variation Distance (Ji et al., ICLR 2023)
>
> [4] Bringing Order to Abstractive Summarization (Liu et al., ACL 2022)
>
> [5] How Far Can Camels Go? Exploring the State of Instruction Tuning on Open Resources (Wang et al., NeurIPS 2023)

---

> > ### Author Response · Authors · 2023-11-21
> > **Comment by Authors**
> >
> > We have considered your valuable suggestions and have made revisions in our manuscript (marked in blue). If you have additional questions or feedback, do not hesitate to reach out to us. If our response has effectively addressed your issue, we would appreciate it if you could consider raising the rating. Thank you!

---

> > ### Comment · Reviewer_p8xo · 2023-11-21
> > **Response to rebuttal**
> >
> > Thank you for the additional details and experiments.
> >
> > I believe that the additional experimental evidence will make the paper somewhat stronger, and am happy to raise my score from 5->6 (as lack of experimental evidence was my main reason for the low rating in the first place). However, I am still not super-convinced that this paper is going to be hugely impactful given the current results, and thus still think it is more-or-less on the borderline.

---

### Official Review · Reviewer_Fsrz · 2023-11-03

**Soundness:** 4 excellent
**Presentation:** 4 excellent
**Contribution:** 3 good
**Rating:** 6
**Confidence:** 4

**Summary:**

This paper introduces a new method for noisy data truncation. It computes the l2 norm of the difference between the model's token distribution and the one-hot groundtruth. The error norm provides a measure of data quality. The Experiments on language modeling, machine translation, and summarization demonstrate the effectiveness of the proposed method.

**Strengths:**

- The proposed method is simple, effective, and robust.
- The analysis experiments are insightful.
- The experiment results on language modeling, machine translation, and summarization are comprehensive and the results are good.

**Weaknesses:**

- The improvements over existing methods seem marginal in some experiments.
- The motivation of the proposed method is a bit unclear to me. See my question below.

**Questions:**

One motivation of the proposed method is that previous works treat all non-ground truth tokens as equally incorrect. However, I do not see how the proposed method solve this problem. In my opinion, all non-ground truth tokens are treated in the same way in the proposed method. can you explain?

Following above, I think one potential improved version of the proposed method is to take into account the similarity between non-ground truth tokens and ground truth tokens when computing the l2 norm.

---

> ### Author Response · Authors · 2023-11-19
> **Response to Reviewer Fsrz**
>
> We sincerely thank the reviewer for the helpful and insightful comments. We appreciate that the reviewer acknowledges that our method is “simple, effective and robust” and our experiments are “insightful” and “comprehensive”.
>
> We address the concerns and questions shared in the review:
>
> > The improvements over existing methods seem marginal in some experiments.
>
> While the improvements may seem marginal, we believe they are meaningful gains for the following reasons:
>
> 1. **Our experimental setup gives a lot of advantage to our baselines:** The baseline results on opus-MT and CNN/DM have already been tuned extensively, and even the state-of-the-art truncation methods [1] can only yield, in your words, “marginal” improvements (+0.58 Rouge-1 and +0.78 BLEU) with exhaustive hyper-parameter tuning (e.g. [1] performed a grid search of 12 combinations of their two hyper-parameters). On the contrary, our proposed approach shows consistent gains in MT (+0.5 BLEU averaged across all directions) with minimal hyper-parameter tuning. We show the number of hyper-parameter configurations we tuned in the following table:
>
> | Method         | Number of Hyper-parameters Tuned |
> |---------------------|----------------------------------|
> | TaiLr [1]           | 10                               |
> | Loss Truncation [2] | 3                                |
> | ENT-Threshold       | 3                                |
> | ENT-Fraction        | 1                                |
>
> 2. **Consistent across different tasks and models:** In spite of the strong baselines, our method yields large improvements in language modeling (-1.58 Perplexity) and machine translation with noise injected (up to +2 BLEU) with no computational overhead. Our method also yields consistent gains on **multiple languages** in multilingual MT and on **two different models** on text summarization. In Appendix F, our method yields larger gains (+0.5 - 0.9 BLEU) on an unfiltered Machine Translation dataset (Paracrawl),
> showing that our method yields larger gains in noisier settings.
>
> > In my opinion, all non-ground truth tokens are treated in the same way in the proposed method. can you explain?
>
> We understand that there is a nuance in how we define “all non-ground truth tokens are treated in the same way”.  We meant that existing methods ignore the predicted distribution over the non-ground truth tokens - therefore treating all probability distributions over the non-ground truth tokens as equal, whereas our method differentiates between when the predicted distribution over non-target tokens is skewed versus uniform. Empirically, we found compelling evidence that the error norm clearly distinguishes between clean and noisy tokens (Figure 4). Thank you for pointing this out, and we have made revisions to clarify our motivation in Section 2 (Marked in blue).
>
> > Following above, I think one potential improved version of the proposed method is to take into account the similarity between non-ground truth tokens and ground truth tokens when computing the l2 norm.
>
> Your point is well taken. We agree that a variant of our method that incorporates semantic similarity is an exciting future direction. However, one challenge is that the target and non-target token could be very similar and the data might still be an error (e.g., The capital of Maryland is Baltimore V.S. Annapolis). As also mentioned by Reviewer p8xo, Figure 4 provides compelling evidence that the error norm itself clearly distinguishes between clean and noisy data, therefore we opted to only use error norm for simplicity.
>
> [1] Tailoring Language Generation with Total Variation Distance (Ji et al., ICLR 2023)
>
> [2] Improved Natural Language Generation via Loss Truncation (Kang & Hashimoto, ACL 2020)

---

> > ### Comment · Reviewer_Fsrz · 2023-11-21
> >
> > The authors addressed my concerns in their rebuttal. I am very happy to learn that they revised the motivation part to avoid confusion. I will keep leaning towards positive.

---

### Official Review · Reviewer_FiyE · 2023-11-22

**Soundness:** 3 good
**Presentation:** 3 good
**Contribution:** 3 good
**Rating:** 6
**Confidence:** 4

**Summary:**

This paper proposes a method called Error Norm Truncation (ENT) to enhance the robustness of text generation models against errors in the training data. The Error Norm Truncation (ENT) method is to truncate training data with high L2 error norm. The comprehensive experiments show that ENT improves the robustness of language models and machine translation models against various types of noise and outperforms previous methods.

**Strengths:**

- This paper studies a fundamental issue in training neural models.
- The proposed method is easy to implement and sounds appealing.
- Experiments show that the proposed method can outperform previous methods on some tasks such as machine translation and language modeling.

**Weaknesses:**

- Even though the results have proven the effectiveness of ENT, I think the motivation of using error norm to estimate data quality can be further discussed and provide more insights about how do they correlate.
 - Theoretically, whether an example is helpful to train a model not only depends on the correctness of its label but also on the uncertainty (or entropy) of this example, according to the lessons from active learning. As uncertainty also takes into account of the prob of non-targets as ENT does. Therefore, it would be important to discuss the relationship between uncertainty and ENT in this paper.
- The proposed method achieves significant improvements on simulated training data with manually added noise (with two types of noise) but it only yields modest improvements on the standard benchmarks where training data may contain less noise. Therefore, it would be helpful if the proposed method works well on natuarally noisy benchmarks. Of course, it may be difficult to collect a large scale of training data and thus it is practical to apply the proposed method under the finetuning scenario, where a small scale of naturally noisy data is used for finetuning (for example, there is such a shared task in WMT).
p.s. I know there is no time for authors to add new experiments into the paper, because I am an emergency reviewer and submit the reviews just before the deadline. However, I would be happy to see more experiments from the dialog box in the openreview system a couple of days later.

**Questions:**

N/A

---

> ### Author Response · Authors · 2023-11-29
> **Author Response to Reviewer FiyE**
>
> We sincerely thank the reviewer for the emergency review and insightful comments. We appreciate that the reviewer acknowledges the significance, simplicity, and effectiveness of our method.
>
> We hope to address the concerns of the reviewer below:
>
> > I think the motivation of using error norm to estimate data quality can be further discussed and provide more insights about how do they correlate.
>
> We show that using the error norm to detect noise is both **theoretically grounded** and **empirically effective**:
>
> - **Theoretical:** The impact on the training loss when a single datapoint $x, y$ is removed is upper bounded by its gradient norm $||\nabla \mathcal{L}(x, y)||$ [1]. Moreover, the impact on test loss when a single datapoint is removed can be also upper bounded by the dot product between its gradient and the gradient on the test dataset $\nabla \mathcal{L}(x, y)^\top \nabla \mathcal{L}(x_\text{test}, y_\text{test})$ [2], indicating that the single sample gradient plays an important role in estimating the data utility. However, single sample gradients are impractical to obtain during batched training. We can approximate the single sample gradient with the error L2 norm [1]. **Therefore, by truncating tokens with a high error l2 norm, we are truncating tokens that have a high impact on the loss, which are likely to be noisy examples.** More discussions on relations with Influence Functions and other data utility measurement methods can be found in Appendix B.
>
> - **Empirical:**  An established understanding of L2 norm is that it is more sensitive to outliers [3] than L1 norm (or equivalently, log-likelihood loss, proof in Appendix A). Therefore, outliers in the data would have larger L2 norms. As also mentioned by reviewer p8xo, Figure 4 provides strong empirical evidence on L2 norm clearly distinguishing between clean and noisy tokens.
>
> If there are any details that need further improvements, we are open to your suggestions.
>
> > Therefore, it would be important to discuss the relationship between uncertainty and ENT in this paper.
>
> Thank you for pointing this out, **we have included extended discussions between ENT and uncertainty sampling in Appendix B**. While both ENT and uncertainty sampling use predicted probabilities to select data, generally, uncertainty sampling for active learning either focuses on low predicted probability (high loss) without considering context entropy or on high entropy without considering probability on the target token. **In contrast, ENT truncates examples with both low predicted probability and low entropy, providing a more accurate estimation of data quality (see Figure 5).** Again, we have added a coverage of the related work for your attention. If there is anything that you’d like us to cover here, please let us know.
>
> > thus it is practical to apply the proposed method under the finetuning scenario
>
> We agree with the reviewer that it is practical to apply ENT on fine-tuning off-the-shelf models. **However, it is also practical to apply ENT to large-scale pre-training of language models as web-crawled data contains various types of noise.** We have shown that our method achieves significant gains in standard language modeling (-1.58 perplexity on Wikitext-103). Moreover, compared to methods that use handcrafted heuristics to filter out noisy data, our method is more general and can be applied during training without additional compute.
>
> > Therefore, it would be helpful if the proposed method works well on naturally noisy benchmarks, …, where a small scale of naturally noisy data is used for finetuning (for example, there is such a shared task in WMT).
>
> In Appendix F, we reported results on Machine Translation trained directly on **naturally noisy Paracrawl data** and evaluated on the WMT 22 test set. Our method achieves +0.5-0.9 BLEU over the baseline on three different language pairs. Since BLEU scores can be imperfect estimators of the true translation quality [5], we also report the COMET scores (Unbabel/wmt-comet22-da) below:
>
> |               | En-Cs         | En-Ru         | En-Zh       |
> |---------------|---------------|---------------|-------------|
> | MLE           | 25.2/69.6     | 24.6/69.5     | 12.5/63.8   |
> | ENT-Threshold | **25.7/70.9** | **25.5/70.2** | **13/65.5** |
>
> **ENT achieves +0.63 BLEU and +1.23 COMET on average over the baseline in naturally noisy settings.**
>
> References:
>
> [1] Deep Learning on a Data Diet: Finding Important Examples Early in Training (Paul et al., NeurIPS 2021)
>
> [2] Understanding Black Box Predictions via Influence Functions (Koh and Liang, ICML 2017)
>
> [3] Chapter 6, Convex Optimization (Boyd and Vandenberghe 2004)
>
> [4] A sequential algorithm for training text classifiers. (Lewis and Gale, SIGIR 1994)
>
> [5] Results of WMT22 Metrics Shared Task: Stop Using BLEU – Neural Metrics Are Better and More Robust (Freitag et al., WMT 2022)

---

### Official Review · Reviewer_CBUj · 2023-11-27

**Soundness:** 3 good
**Presentation:** 3 good
**Contribution:** 3 good
**Rating:** 6
**Confidence:** 4

**Summary:**

This work proposes Error Norm Truncation (ENT), a method to clean low-quality data in training text generation models. The proposed method measures the distribution of each token between the predicted distribution and the ground-truth distribution, which is more robust than previous works of measearuing loss or probability.

**Strengths:**

1. The first half of the paper is well-organized and clearly written. I enjoy reading this part, which provides me a clear understanding of your motivation and method.
2. The proposed method is technically sound, echoing your motivation, with theoritical proof and experimental studies.
3. The experiments, overall, provide a good evidence of the effectivenss of the proposed method.

**Weaknesses:**

1. The last half of the paper is not well wrtitten with some temrs not clearly explained:
1.1 In Figure 5, ENT (largest) and ENT (smallest) are not explained. I did not find where explains these two terms.
1.2 In Figure 6, what and how are the ENT threshold and ENT fraction set?
2. The authors claim that previous work would rely on hyper-parameter tuning (on fraction or threshold). But the proposed ENT method also requires these hyper-parameters. I am not sure the proposed method would be more efficient on hyper-parameter selecting.
3. In $5.2, apart from the two settings, another setting "Over translated" (noted by the authros in Figure 2) should also be included.
4. In $5.4, it is inadequate to draw such a conclusion on using which method for pre-trained model and using which method for from-sractch model, with only a summarization task. More downstream tasks should be conducted to draw such a conclusion.
5. Title of Section 4 could be re-considered to like "Analysis", as case studies always provide study on a few specific "cases" from the dataset. But this section is analyzing on the whole dataset. However, this is just a suggestion, and does not affect my rating.

**Questions:**

1. In Table 1, when the untranslated ration is 10%, MLE outputperforms all the methods with a relatively large margin. This is interesting and also confusing, comparing to other ratio. Do you have any explain or study on this? Moreover, in real scenoria, I think the untranslated ratio in the dataset is not as large as more than 20%, so does this result suggest that under common situation, using MLE is enough to beat other modifications?
2. In Table 2, the advantages of ENT and also the Loss Trunc. and TaiLr are not significant against MLE when ratio <=30%. Also in real scenoria, the misordered ratio might not be that high, which also raieses the same question as above.

---

> ### Author Response · Authors · 2023-12-02
> **Author Response to Reviewer CBUj**
>
> We sincerely thank the reviewer for the thoughtful comments and suggestions. We are grateful that the reviewer finds our paper clearly written, well-motivated, and empirically sound.
>
> We hope to address the concerns of the reviewer below:
>
> > 1.1 In Figure 5, ENT (largest) and ENT (smallest) are not explained. I did not find where explains these two terms.
>
> ENT (largest) refers to truncating the sentences with largest error norm, ENT (smallest) refers to truncating sentences with smallest ENT. Thank you for pointing this out. We have added a footnote to clarify (marked in blue).
>
> > 1.2 In Figure 6, what and how are the ENT threshold and ENT fraction set?
>
> We choose the best of {1.35, 1.38, 1.4} for ENT-Threshold and use fraction = 0.1 for ENT-fraction in all our experiments. More details on hyper-parameters can be found in Appendix C.  For figure 6, we additionally varied the starting iteration in (0, 100, 200, 500, 100) at which we start to apply data truncation methods (ENT, Loss Trunc, and TaiLr). We have made annotations in Figure 6 to clarify.
>
> > The authors claim that previous work would rely on hyper-parameter tuning (on fraction or threshold). But the proposed ENT method also requires these hyper-parameters. I am not sure the proposed method would be more efficient on hyper-parameter selecting.
>
> We report the number of hyper-parameters we tuned for each baseline below:
>
> | Method         | Number of Hyper-parameters Tuned |
> |---------------------|----------------------------------|
> | TaiLr [1]           | 10                               |
> | Loss Truncation [2] | 3                                |
> | ENT-Threshold       | 3                                |
> | ENT-Fraction        | 1                                |
>
> In addition to fraction/threshold, the starting iteration at which we start to apply other baselines can have a significant impact on the final performance, as shown in Figure 6. **In contrast, ENT exhibits the best performance and least variance across different starting iterations.**
>
> > In $5.2, apart from the two settings, another setting "Over translated" (noted by the authors in Figure 2) should also be included.
>
> We included experiments by adding random words into the target sentence in opus En-Fr:
>
> |     | 0%   | 10%  | 20%  | 30%  | 40%  | 50%  |
> |-----|------|------|------|------|------|------|
> | MLE | 36.5 | 36.3 | 36.1 | 36.0 | 35.8 | 35.4 |
> | ENT-Fraction | 36.7 | 36.7 | 36.6 | 36.4 | 36.3 | 36.1 |
>
> Our results show that adding random words into the target sentence does not harm the performance as much as untranslated target sentences, and **ENT was able to outperform the MLE baseline at all noise levels.**
>
> > In $5.4, it is inadequate to draw such a conclusion on using which method for pre-trained model and using which method for from-sractch model, with only a summarization task. More downstream tasks should be conducted to draw such a conclusion.
>
> We thank the reviewer for pointing this out. We have revised this section in our latest manuscript.
>
> > In Table 1, when the untranslated ration is 10%, MLE outputperforms all the methods with a relatively large margin. This is interesting and also confusing, comparing to other ratio. Do you have any explain or study on this?
>
> We suspect that the superior performance of MLE at small noise ratios can be attributed to certain engineering choices in the baseline. For example, existing work has shown that label smoothing and the choice of optimizers can have influence on robustness [1,2]. We agree that quantifying the strength of implicit robustness enhancement effect of label smoothing and optimizers, as well as how they interact with data truncation methods is an exciting direction for further research.
>
> > so does this result suggest that under common situation, using MLE is enough to beat other modifications?
>
> While MLE outperforms others when 10% of the entire data is injected, we wouldn't conclude that MLE is enough to beat data truncation methods under a common situation. In practice, **the type and the amount of noise can vary depending on the dataset**, especially if we take the possibility of an adversary manually injecting sentences with multiple types of noise into the training corpus.
>
> Under a more practical setting, we validate our method on **naturally noisy Paracrawl data** and report the BLEU/COMET scores the WMT 22 test set:
>
> |               | En-Cs         | En-Ru         | En-Zh       |
> |---------------|---------------|---------------|-------------|
> | MLE           | 25.2/69.6     | 24.6/69.5     | 12.5/63.8   |
> | ENT-Threshold | **25.7/70.9** | **25.5/70.2** | **13/65.5** |
>
> **ENT achieves +0.63 BLEU and +1.23 COMET on average over the baseline in a practical, naturally noisy setting.**
>
> [1] In and Out-of-Domain Text Adversarial Robustness via Label Smoothing (Yang et al., ACL 2023)
>
> [2] Understanding the robustness of difference between stochastic gradient descent and adaptive gradient methods. (Ma et al., 2023)

---

### Meta-Review · Area_Chair_2reR · 2023-12-10

**Metareview:**

This work proposes Error Norm Truncation (ENT), a method to clean low-quality data in training text generation models. The proposed method measures the distribution of each token between the predicted distribution and the ground-truth distribution, which is more robust than previous works of measuring loss or probability.

All reviewers admit that it is a fundamental problem and this paper proposes a nice way to solve it. It is a clear acceptacne.

**Justification For Why Not Higher Score:**

n/a

**Justification For Why Not Lower Score:**

n/a

---

### Decision · Program_Chairs · 2024-01-16

Accept (spotlight)